# The cGAS/STING/IFN-1 Response in Squamous Head and Neck Cancer Cells after Genotoxic Challenges and Abrogation of the ATR-Chk1 and Fanconi Anemia Axis

**DOI:** 10.3390/ijms241914900

**Published:** 2023-10-04

**Authors:** Sebastian Zahnreich, Soumia El Guerzyfy, Justus Kaufmann, Heinz Schmidberger

**Affiliations:** Department of Radiation Oncology and Radiation Therapy, University Medical Centre of the Johannes Gutenberg, University Mainz, 55131 Mainz, Germanyheinz.schmidberger@unimedizin-mainz.de (H.S.)

**Keywords:** head and neck cancer, ionizing radiation, cisplatin, cGAS, STING, type 1 interferons, cytosolic DNA, ATR, Chk1, Fanconi anemia

## Abstract

Locally advanced head and neck squamous cell carcinomas (HNSCC) are often refractory to platinum-based radiochemotherapy and new immuno-oncological strategies. To stimulate immunogenic antitumor responses in HNSCC patients, we investigated the cGAS/STING/IFN-1 signaling pathway after genotoxic treatments and concomitant abrogation of the DNA damage response (DDR). For this purpose, FaDu and UM-SCC1 cells were exposed to X-rays or cisplatin and treated with an ATR or Chk1 inhibitor, or by Fanconi anemia gene A knockout (FANCA ko). We assessed clonogenic survival, cell cycle regulation, micronuclei, free cytosolic double-stranded DNA, and the protein expression and activity of the cGAS/STING/IFN-1 pathway and related players. Cell survival, regulation of G2/M arrest, and formation of rupture-prone cGAS-positive micronuclei after genotoxic treatments were most affected by ATR inhibition and FANCA ko. In UM-SCC-1 cells only, 8 Gy X-rays promoted IFN-1 expression unaltered by abrogation of the DDR or concomitant increased TREX1 expression. At a higher dose of 20 Gy, this effect was observed only for concurrent Chk1- or ATR-inhibition. FANCA ko or cisplatin treatment was ineffective in this regard. Our observations open new perspectives for the enhancement of cGAS/STING/IFN-1-mediated antitumor immune response in HNSCC by hypofractionated or stereotactic radiotherapy concepts in multimodal settings with immuno-oncological strategies.

## 1. Introduction

Head and neck squamous cell carcinoma (HNSCC) was the eighth most common cancer worldwide with 878,348 new cases and 444,347 deaths in 2020 [1]. About 60% of HNSCC patients present with locally advanced (LA) stage III or IV disease, characterized by frequent resistance to current treatment standards, locoregional recurrence, and metastasis causing dismal 5-year overall survival rates below 50% [2,3]. The standard of care for LA HNSCC patients has remained virtually unchanged over the past two decades as surgery with adjuvant platin-based chemoradiotherapy (CRT) for operable tumors or definitive CRT for unresectable tumors [4,5]. About 50% of patients show recurrence after CRT and, in patients with contradictions for cisplatin (CDDP), definitive radiotherapy (RT) has even poorer outcomes [6,7]. Other multimodal strategies with taxanes, antifolates, and cetuximab are used only in palliative settings with response rates usually below 30% [8]. Currently, immuno-oncological approaches exploit the antitumor effects of immune checkpoint inhibitors (ICI) targeting programmed cell death protein 1 (PD-1) and its ligand PD-L1 or cytotoxic T-lymphocyte-associated protein 4 (CTLA-4) for HNSCC patients. However, so far, durable response rates are low and outcomes are generally not superior to the current standards of care for LA HNSCCs [9,10,11,12,13]. For these patients, there remains an unmet medical need for enhancing the therapeutic efficacy of ICI.

Currently, new irradiation concepts for RT–ICI combinations are being developed for the treatment of HNSCC to reduce the immunosuppressive effect of RT caused by the inevitable irradiation of the hematologic system [14,15]. In particular, stereotactic body radiotherapy (SBRT) based on hypofractionated irradiation of small volumes with doses of >5 Gy can achieve sparing of tumor-draining lymph nodes and peripheral blood to elicit therapeutically relevant anticancer immune responses of ICI.

The antitumor immune response of cancer patients can also be stimulated by the genotoxic impact of RT or chemotherapy (CT) on tumor cells to turn non-inflamed (cold) into inflamed (hot) tumors [16]. An increase in the immunological infiltrate in the tumor microenvironment can be promoted through the release of damage-associated molecular patterns (DAMPs) by immunogenic cell death, chemokine secretion, and the immunostimulatory effect of the type-I interferon (IFN-1) pathway mediated by the cyclic GMP-AMP (cGAMP) synthase (cGAS) and its downstream adaptor stimulator of interferon genes (STING) [17,18,19]. The binding of 2′,3′-cGAMP to STING triggers the phosphorylation of interferon regulatory factor 3 (IRF3) via TANK binding kinase 1 (TBK1) leading to the transcriptional activation of inflammatory genes, including IFNs.

cGAS is activated by free cytosolic double-stranded DNA (cdsDNA) or the formation and rupture of micronuclei (MN) loaded with DNA fragments, which are potently induced by the clastogenic effects of ionizing radiation (IR) or certain cytostatic drugs [18]. Sensitizing strategies through inhibition of the DNA damage response (DDR) may boost the cGAS/STING/IFN-1 signaling cascade and antitumor immune effects [20]. cGAS/STING signaling and IFN-1 expression in tumor cells after hypofractionated IR exposure have been associated with sustained immunogenic locoregional and distant (abscopal) antitumor responses in the context of an immune checkpoint blockade [17]. In HNSCCs, STING expression also correlates with increased immune infiltration of the tumor microenvironment and is a favorable prognostic factor [21]. However, patient benefits often remain limited since cGAS/STING signaling is silenced in many tumor entities or antagonized, e.g., by the human papillomavirus (HPV)-associated oncoprotein E7 in HPV16-positive oropharyngeal carcinomas [21,22,23,24]. cGAS/STING signaling is also counteracted by increased expression of three prime repair exonuclease 1 (TREX1), degrading cdsDNA after high ablative single doses of IR, commonly exceeding 10 Gy of photons [17]. In a limited number of HPV-negative STING-expressing HNSCC cell lines, a general activation of the cGAS/STING/IFN-1 pathway by DNA transfection has previously been demonstrated but with cell-line-dependent variability [22,24]. However, the response of this signaling cascade to DNA-damaging cancer therapies alone or with concomitant sensitizing strategies by pharmacological inhibitors of the DDR has hitherto not been explored in HNSCCs.

Also, intrinsic deactivation of the DDR by genetic defects, as in rare DNA repair deficiency syndromes like Fanconi anemia (FA) or BRCAness in hereditary breast and ovarian cancer, triggers cGAS/STING signaling and inflammatory processes through genomic instability [25,26]. Somatic mutations, hypermethylation, and copy number variations of FA/BRCA repair pathway genes have also been described in more than 30% of sporadic HPV-negative HNSCCs associated with early onset and poor prognosis [27,28,29,30,31,32]. On the other hand, a compromised FA/BRCA repair pathway fosters sensitization to DNA-damaging tumor therapies and a pro-inflammatory phenotype including upregulation of STING expression [27,33,34]. FA patients themselves have a several-hundred-fold increased risk of developing aggressive HNSCC in early adulthood compared with the general population [33,35,36]. Fatally, these patients tolerate DNA-damaging cancer therapies poorly, leading to significant morbidities, and non-genotoxic treatment options including immuno-oncological strategies are urgently needed.

To address these clinically relevant issues and improve treatment outcomes of sporadic HNSCCs and in FA, we investigated stimulation of the cGAS/STING/IFN-1 pathway by therapy-relevant genotoxic challenges and concomitant impairment of the DDR. For this purpose, two HNSCC cell lines were examined in this regard after exposure to IR or CDDP and pharmacological inhibition of DNA damage response kinases ataxia telangiectasia and Rad3-related (ATR) and checkpoint kinase 1 (Chk1) or gene knockout of FA complementation group A (FANCA ko).

## 2. Results

### 2.1. Clonogenic Survival

The cytotoxic effects of the different treatments in HNSCC cells were evaluated using a clonogenic survival assay (Figure 1). For exposure to X-rays or CDDP alone, a dose-dependent reduction in cell survival was observed for both HNSCC cell lines. The response after IR was comparable between both HNSCC cell lines, while UM-SCC-1 cells were significantly more sensitive to CDDP compared to FaDu cells (*p* < 0.05). For X-rays, inhibition of ATR resulted in the greatest reduction in cell survival, followed by FANCA ko. Inhibition of Chk1 did not reduce cell survival after IR or CDDP and even resulted in a significant increase at 2 µM CDDP compared with CDDP alone. After CDDP treatment, FANCA ko resulted in the strongest cell inactivation, followed by inhibition of ATR. The overall trends in survival rates for the various treatments were comparable between the two HNSCC cell lines with different p53 status.

### 2.2. Cell Cycle and G2/M Checkpoint Abrogation

Cell cycle progression through mitosis after genotoxic impacts is a prerequisite for the formation of MN or free cdsDNA to activate cGAS/STING signaling [37]. The effects of the different treatments on cell cycle regulation with a focus on abrogation of the DNA-damage-induced G2/M checkpoint by the ATR inhibitor (ATRi) and Chk1 inhibitor (Chk1i) were investigated by flow cytometric cell cycle analysis (Figure 2).

In p53-mutated FaDu cells, IR exposure caused a significant increase in the fraction of G2 cells compared with unirradiated cells (*p* < 0.001) indicating a potent induction of G2/M arrest (Figure 2A). Concurrent treatment with IR and Chk1i or ATRi resulted in mild but significant or complete abrogation of the IR-induced G2/M arrest, respectively. In FaDu FANCA ko cells, the IR-induced G2/M arrest was more pronounced than in FaDu wt cells due to the sole DNA repair defect. In p53-non-mutated UM-SCC-1 cells, there was only a weak and non-significant increase in the fraction of G2/M cells after IR exposure alone compared with unirradiated cells (Figure 2B). However, treatment with IR plus Chk1i or ATRi resulted in a comparable, nonsignificant decrease in the G2/M fraction to the level of unirradiated cells. In contrast with FaDu FANCA ko cells, UM-SCC-1 FANCA ko cells already showed a significant increase in the fraction of G2/M cells in unirradiated populations with no change after IR exposure.

The treatment with the DNA cross-linking drug CDDP for 24 h resulted in a significant accumulation of cells in the S-phase in both HNSCC cell lines due to replication stress (*p* < 0.001) (Figure 2C,D). Inhibition of the DDR had no significant effect on cell cycle distributions or progression through the S-phase during CDDP treatment. Only Chk1i-treated FaDu cells showed a weak but non-significant reduction of the CDDP-induced proportion of cells in the S-phase.

Together, treatment with the Chk1i, but most pronounced with the ATRi, abrogated IR-induced G2/M arrest and allowed cell cycle progression through mitosis.

### 2.3. Induction of Micronuclei

MN are potently induced by clastogenic agents and trigger the cell-intrinsic autoinflammatory cGAS/STING signaling cascade [38]. As shown in Figure 3, the ATRi or FANCA ko increased the level of basal MN in both HNSCC cell lines due to compromised repair of replication-associated DNA damage. FaDu cells showed overall higher numbers of MN compared with UM-SCC-1 cells. The rate of sporadic MN was about three-fold higher for untreated FaDu wt cells than UM-SCC-1 wt cells (*p* < 0.05), and this observation also applied to the genotoxic treatments. IR exposure resulted in the most significant increment in MN frequencies compared with unirradiated populations in both FaDu (*p* < 0.001) and UM-SCC-1 (*p* < 0.05) cells. Again, the amount of IR-induced MN in FaDu wt cells was 2.6-fold higher than in UM-SCC1 wt cells (*p* < 0.05). Only inhibition of ATR in FaDu cells resulted in a significant 2.4-fold increase in the yield of IR-induced MN compared with irradiated wt cells. Treatment with CDDP alone did not significantly elevate the numbers of MN, only by concomitant inhibition of ATR in both cell lines and for UM-SCC-1 FANCA ko cells. In particular, for CDDP plus ATRi, MN frequencies were elevated six-fold for FaDu compared with UM-SCC-1 cells (*p* < 0.01). The MN data revealed significant variations in the overall level between the two HNSCC cell lines, and the most significant increases occurred after exposure to IR and by inhibition of ATR for both genotoxic treatments.

To establish a link between the formation of MN representing membrane-bound cytosolic DNA fragments and activation of the cGAS/STING pathway in HNSCC cells, we examined the integrity of MN and the recruitment of cGAS using immunolabeling and fluorescence microscopy in a modified cytokinesis block micronucleus (CBMN) assay (Figure 3C,D). First, the proneness to rupture and permeability of MN in HNSCC cells was investigated by performing immunostaining with a monoclonal antibody against dsDNA using a mild permeabilization protocol that did not affect the nuclear envelope. As shown in Figure 3C, strong staining of dsDNA in MN or chromatin bridges was detected, indicating the high fragility and permeability of MN in HNSCC cells. Next, we performed immunolabeling of cGAS using a conventional permeabilization protocol and observed strong accumulation and hyperintense signals of cGAS restricted to MN in both HNSCC cell lines (Figure 3D). Our results indicate potent activation of the cGAS/STING signaling pathway by MN in HNSCC cells and enhancement by genotoxic treatments.

### 2.4. cGAS/STING/IFN-1 Response

Based on the observations of the generation of MN with fragile envelopes and strong cGAS recruitment in both HNSCC cell lines, protein expression and activation of the key players of the cGAS/STING signaling cascade were examined by Western blot. For the exposure to IR, we adopted single doses relevant to RT of 2 Gy for normofractionated regimens and high doses of 8 Gy and 20 Gy with relevance for hypofractionated regimens and SBRT. First, kinetics at 24, 48, and 72 h after exposure to X-rays or 2 µM CDDP alone were examined in HNSCC wt cells. The corresponding Western blots are shown in Figure 4A,B, and the densitometric quantification of the signals can be found in Appendix A. In FaDu wt cells, the only pronounced alteration was an increase in the Tyr701-phosphorylated isoform of STAT1 after high doses of X-rays or CDDP treatment, which can be stimulated by IFN-1 [39]. However, increased IFN-β protein expression was not observed in FaDu wt cells after the respective exposures. In contrast, more pronounced responses were observed in UM-SCC1 wt cells after IR. We found increased levels of STING, pIRF3, pSTAT1, and especially IFN-β expression 72 h after exposure to 8 Gy. After treatment of UM-SCC1 wt cells with CDDP, the expression of most proteins was downregulated or did not differ from untreated cells. Based on these results, we focused on the time point of 72 h post-treatment for a detailed investigation of a cGAS/STING/IFN-1 response after genotoxic exposures with the simultaneous abrogation of the DDR. Representative Western bots are shown in Figure 4C,D; all Western blots and densitometric quantification of signals are provided in Appendix A.

In FaDu wt cells, the inhibition of the ATR-Chk1 axis did not stimulate the cGAS/STING/IFN-1 response 72 h after genotoxic exposures. However, FANCA ko promoted a marked increase in STING expression, which was not accompanied by increased activity of its downstream targets or IFN-β expression supporting the previous notion of a general inactivity of this pathway in FaDu cells.

In UM-SCC-1 wt cells, the most striking response was again the greatly increased expression of IFN-β 72 h after exposure to 8 Gy X-rays, which was unaffected by concomitant treatment with the Chk1i or ATRi. Contrarily, after exposure to 20 Gy, a significant increase in IFN-β expression was observed only by inhibition of Chk1 and, to a lesser extent, by inhibition of ATR. Strikingly, the same expression patterns were observed for the DNA exonuclease TREX1, which is expected to antagonize the cGAS/STING signaling pathway and an associated IFN-1 expression by degrading cytosolic DNA fragments. However, the expression and activation levels of the target proteins of the cGAS/STING signaling cascade after IR exposure were less pronounced and occurred mainly for elevated levels of STING after 8 Gy or pIRF3 after IR in general. However, such patterns did not emerge as consistently as for IFN-β and TREX1 protein levels. Also, an increase in PD-L1 expression was observed after IR in UM-SCC-1 wt cells, which was more pronounced after 2 Gy and 8 Gy (+ATRi) than after 20 Gy. FANCA ko in UM-SCC-1 cells resulted in a general downregulation of the cGAS/STING cascade, PD-L1, as well as TREX1. Also, CDDP treatment of UM-SCC-1 cells caused an overall decrease in protein levels with a concomitant increase in phosphorylated isoforms, particularly for STING, IRF3, and STAT1. This, however, was not associated with an increased expression of IFN-β. Also, TREX1 expression was not stimulated by CDDP treatment whereas PD-L1 expression increased. Inhibitors of the DDR or the FANCA ko had no noticeable effect on CDDP-related patterns of protein expression and activation.

### 2.5. cdsDNA

Finally, to elucidate a relationship between the presence of free cdsDNA after genotoxic challenges and the observed alterations in protein levels and activity status of the cGAS/STING/IFN-1 pathway shown in Figure 4, free cdsDNA was extracted and quantified 72 h after the same treatment conditions in HNSCC cells (Figure 5).

As for the quantification of MN, the amounts of cdsDNA were consistently higher in FaDu cells than in UM-SCC-1 cells. After exposure to IR, dose-dependent increases in free cdsDNA were observed, which were significantly elevated in FaDu cells only after 8 Gy and 20 Gy for all treatments compared with unirradiated cells (*p* < 0.05). In UM-SCC-1 cells, a significant increase over unirradiated cells occurred only at the very high dose of 20 Gy for ATRi treatment (*p* < 0.001) or the FANCA ko (*p* < 0.001). Impairment of the DDR did not significantly increase IR-induced amounts of cdsDNA, except for FANCA ko in UM-SCC-1 cells after 8 Gy and 20 Gy. Treatment with 2 µM CDDP significantly increased the amounts of cdsDNA only for FaDu cells for wt (*p* < 0.05) and FANCA ko (*p* < 0.001). As for IR exposure, impairment of the DDR did not significantly increase CDDP-induced amounts of cdsDNA, except for FNACA ko in UM-SCC-1 cells (*p* < 0.05).

## 3. Discussion

In LA HNSCCs, treatment outcomes of CRT as the standard of care remain poor and have not been significantly improved by the implementation of immuno-oncologic strategies in multimodal settings so far. However, DNA-damaging cancer therapies can support both local and distant tumor immunogenicity through activation of the cGAS/STING/IFN-1 pathway in tumor cells [17]. Since these effects are virtually unexplored in HNSCC, we investigated the impact of IR and CDDP with concomitant impairment of the DDR by inhibition of the ATR-Chk1 axis or knockout of the FA repair pathway on cGAS/STING/IFN-1 signaling in two HNSCC cell lines. We demonstrated that exposure to a single dose of 8 Gy of X-rays alone strongly enhanced IFN-1 protein expression in the HNSCC cell line UM-SCC-1 after 72 h with no increase due to the abrogation of the DDR. In contrast, for a higher ablative X-ray dose of 20 Gy, inhibition of ATR or Chk1 was necessary to induce this IR-associated immunostimulatory effect. CDDP was generally ineffective in this regard. However, we observed no overall correlation of these results with the level of IR-induced free cytosolic DNA or MN. Also, IFN-1 expression was not antagonized by concomitantly increased amounts of the DNA exonuclease TREX1.

Genotoxic cancer therapies like RT eradicate tumor cells primarily by DNA damage and chromosomal aberrations that lead to mitotic catastrophes generating cytosolic DNA. The latter is present as free cdsDNA or in MN, both of which trigger the immunogenic cGAS/STING/IFN-1 pathway supporting the local and abscopal action of immuno-oncological therapies [40]. Therefore, we first examined the effects of abrogating the DDR on classic cellular endpoints of genotoxic challenges including cell inactivation, cell cycle control, DNA damage, and cytosolic DNA. Inhibition of ATR had the most pronounced effect on cell survival, overriding an IR-induced G2/M arrest, and induction of MN. In contrast, inhibition of the checkpoint kinase Chk1, a downstream target of ATR, had no impact except for a weak abolition of IR-induced G2/M arrest. Previous studies have described sensitizing effects for Chk1 inhibitors after exposure to IR or CDDP in HNSCC cell lines in terms of cytotoxicity and G2/M checkpoint abrogation. However, these effects were mostly not very pronounced and varied within the cell panels studied [41,42]. At the Chk1i concentration used in the present study, we previously showed a potent abrogation of a replication stress-induced G2/M arrest and paradoxical ATR-dependent Chk1 phosphorylation in primary fibroblasts [43]. As expected, disruption of the FA repair pathway with a major role in DNA crosslink repair resulted in a mild radio- but the strongest chemosensitization. The combination of DNA-damaging therapies and inhibitors of the DDR is already being investigated for HNSCCs in clinical trials (e.g., NCT03022409: ceralasertib (ATR), NCT02567422: berzosertib (ATR), NCT04576091: elimusertib (ATR), or NCT02555644 and NCT02555644: prexasertib (Chk1)); however, so far, only concerning tumor cytotoxicity as the major endpoint, e.g., through mechanisms of reproductive cell death or apoptosis [44,45,46]. The immunogenic effect of the cGAS/STING/IFN-1 cascade has not yet been investigated in this regard in HNSCC.

In the present study, we have demonstrated for the first time an IR-induced IFN-1 response in human HNSCC cells 72 h after a single exposure with 8 Gy X-rays. Inhibition of DNA repair and cell cycle regulation did not increase the IFN-1 response at this dose but was necessary to elicit this IR-associated reaction at a higher dose of 20 Gy. The strong impact of the Chk1i at this dosage was surprising as the inhibition of this kinase proved to be largely inefficient at the other endpoints as mentioned above. Nevertheless, our results support the concept that abrogation of the G2/M arrest promotes mitotic catastrophes that favor the formation of cytosolic nucleic acids and trigger a cGAS/STING/IFN-1 response. Since we observed no effect of inhibition of ATR or Chk1 on IFN-1 expression 72 h after irradiation with 8 Gy of X-rays, IR-induced cell cycle arrest may no longer play a role at this dose level but it does at the higher dose of 20 Gy. However, the detailed mechanistic background for the high efficacy of the Chk1i to promote an IFN-1 response at the high dose of 20 Gy in HNSCC cells was not unraveled in this study.

Our data support promising strategies for the combination of ICI with RT techniques that use high ablative photon doses or particles with enhanced relative biological effectiveness, such as SBRT or C-ions, respectively [47]. Hypofractionated and stereotactic RT modalities are gaining clinical importance in HNSCC to preserve immune fitness for RT–ICI combinations compared with conventional normofractionated RT with single tumor doses in the range of 2 Gy [14,15]. Besides the different effects of these RT modalities on the immune landscape of the tumor microenvironment, stereotactic RT techniques can spare both circulating lymphocytes and tumor-draining lymph nodes. This reduces leukopenia and allows dendritic cell priming and T-cell activation to optimize the antitumor immune response, which is supported by IFN-1 expression from irradiated tumor cells [14,15,17]. Thus, the stimulation of the cGAS/STING/IFN-1 pathway by higher radiation doses in HNSCC observed in our study fits very well with the current concepts of optimizing radiation dose and fractionation regimens for the highest efficacy of RT–ICI combinations.

Vanpouille-Box et al. [15] demonstrated that in mouse mammary carcinomas refractory to ICI, hypofractionated RT with 3 × 8 Gy caused cGAS/STING activation and IFN release in tumor cells. The combination with ICI resulted in a very efficient locoregional and distant antitumor response. However, the authors also showed that higher single tumor doses of more than 12–18 Gy induced expression of the DNA exonuclease TREX1, which abolished cGAS/STING/IFN-1-mediated immunogenicity by degrading cytosolic DNA. We observed an increase in TREX1 protein expression in UM-SCC-1 cells already at a dose of 8 Gy. Surprisingly, TREX1 protein levels showed the same expression pattern as observed for IFN-ß, i.e., IFN-1 production was not suppressed by TREX1 in HNSCC cells. Potentially, other signaling pathways of cytosolic RNA sensing such as pattern recognition receptors including RIG-I-like or Toll-like receptors [48] may trigger an IFN-1 response in HNSCC. Recently, Feng et al. [49] showed that at a radiation dose of 20 Gy, concomitant administration of an ATRi (AZD6738) promoted an IFN-1 response in breast, colon, and lung carcinoma cell lines. This was triggered mainly through the RIG-I/MAVS/TBK1-dependent cytosolic RNA sensing pathway by the conversion of AT-rich DNA to RNA. This finding is also of relevance to the present study since we did not observe a consistent pattern of activation of the cGAS/STING pathway, e.g., phosphorylation of TBK1, STAT1, or STING, in UM-SCC-1 cells in the presence of a strong IFN-1 response. The role of these alternative nucleic acid sensing pathways for a therapeutically relevant IFN-1 response should also be considered and investigated in HNSCC.

As another antagonist of the antitumoral immune response and an important target of ICI, we investigated the expression of PD-L1 in HNSCC cell lines. PD-L1 expression on tumor cells can be triggered by genotoxic therapies and promotes tumor immune escape [50,51]. In UM-SCC-1 but not FaDu cells, exposure to IR or CDDP resulted in increased expression of PD-L1. Recently, Sato et al. [52] showed that in tumor cells after exposure to 10 Gy of X-rays, PD-L1 is upregulated by the activity of the DDR kinases ATM, ATR, and Chk1. However, in our study, except for a downregulation due to FANCA ko, we did not detect any effect of inhibiting ATR or Chk1 in PD-L1 protein levels after exposure to IR.

It must be considered that an IR-associated INF-1 response was achieved in only one of the two HNSCC cell lines studied. A lack of IFN-1 response in FaDu relative to UM-SCC-1 cells was previously also reported by Bortnik et al. [24] after transfection with calf thymus DNA despite a robust cGAS and STING protein expression. This difference in stimulability of IFN-1 response in STING-expressing HNSCC cells was discussed with their different anatomic origin: floor of the mouth of the oropharynx for UM-SCC-1 vs. oropharyngeal for FaDu. However, Luo et al. [21] were able to force increased expression of IFN-β in FaDu cells transfected with a STING-expressing plasmid. In our study, elevated expression of intrinsic STING in FaDu cells by FANCA ko did not activate this signaling cascade, suggesting a mutational deactivation of STING in this HNSCC cell line. Another explanation for the different IFN-1 responses between FaDu and UM-SCC-1 cells could be their divergent p53 status. Recent data suggest that p53 plays an important role in the modulation of anti-tumor immunity by engaging the cGAS/STING/IFN-1 pathway [53]. Wild-type p53 can upregulate STING expression, whereas mutational p53 disrupts the cGAS/STING/IFN-1 pathway by preventing TBK1-dependent activation of IRF3 [54,55]. Thus, in HNSCC, p53 status may also play a crucial role in the IFN-mediated antitumoral immune response. In general, the current picture of STING expression and pathway activity in HNSCCs is very heterogeneous. Baird et al. [56] initially attributed STING expression and activity in HNSCC to HPV status and the respective origin of the neoplastic cell population: STING expression in HPV-positive HNSCCs was related to their development from the STING-positive basaloid squamous epithelium, whereas HPV-negative HNSCCs arise from STING-negative differentiated keratinocytes of the epithelium. However, subsequent studies have shown a general variation of STING expression in HPV-positive as well as HPV-negative HNSCC cell lines [21,22,24]. We also examined the protein levels of key players of the cGAS/STING pathway in a panel of seven HPV-negative HNSCC cell lines (Appendix A). The protein expression of STING, cGAS, IRF3, and TBK1 varied considerably, regardless of the anatomic origin of the cells. Surprisingly, Baird et al. [56] showed a STING agonist-mediated tumor regression in an immunocompetent mouse model of STING-negative squamous cell carcinoma. This observation was attributed to the activation of STING and the production of effectors such as TNFα and IFN-1 in STING-expressing non-cancer cells in the tumor stroma. Therefore, an investigation of the IR-associated activation of the cGAS/STING/IFN-1 cascade in non-cancer cells in the tumor microenvironment is also of great interest to support the immunogenic impact of RT in STING-negative or -inactive tumors.

Also, the abrogation of the DDR alone did not stimulate cGAS/STING/IFN-1 signaling in HNSCC cells, as previously described for other tumor entities [57]. Regardless, inhibition of ATR and FANCA ko promoted basal genomic instability. This is commonly discussed with increased STING expression and mediation of a proinflammatory phenotype in syndromes with DNA repair deficiencies including FA [26]. This effect could support the action of immuno-oncologic therapies in FA-associated HNSCC occurring spontaneously or in FA patients [58]. Except for a strong increase in STING expression in FaDu FANCA ko cells, which, however, was not accompanied by an increase in its downstream activity and an IFN-1 response, the FANCA ko rather led to a downregulation of the general protein levels of the cGAS/STING pathway, as well as of TREX1 and PD-L1 in both HNSCC cell lines. Thus, our results differ from the observations that intrinsic DNA repair deficiencies generally support a pro-inflammatory phenotype [26]. However, this is mainly true for normal tissue cells, whereas tumor cells may be regulated differently in this respect due to their mutational background.

Together, the present study demonstrates the activation of the immunogenic cGAS/STING/IFN-1 pathway in HNSCC cells by IR and the need for abrogation of the DNA damage response at very high ablative radiation doses to promote this effect. Due to the poor therapeutic response of HNSCC to immuno-oncologic therapies, synergistic effects with the immunostimulatory impact of conventional DNA damaging therapies are highly warranted to optimize RT–ICI combinations. Our results are particularly relevant in the context of multimodal hypofractionated or stereotactic RT–ICI concepts for HNSCC. However, the overall expression and activity of the cGAS/STING/IFN-1 pathway were variable in our study. We aim to thoroughly characterize and validate this antitumor response in future studies in larger cell panels, tumor biopsies, and preclinical models as a predictive biomarker and therapeutic target in clinical practice for HNSCC.

## 4. Materials and Methods

### 4.1. Cell Culture

Experiments were performed using the HPV-negative HNSCC cell lines FaDu (hypopharyngeal, p53 mutated [59]) and UM-SCC-1 (oral cavity/mouth base, p53 non-mutated [60]). For both cell lines, a CRISPR-Cas9 FANCA ko was used in addition to the wild type (wt). The FA phenotype was confirmed by treatment with the DNA crosslinker Mitomycin C (Appendix A). All cells were provided by the Fanconi cancer cell line repository (https://apps.ohsu.edu/research/fanconi-anemia/cell_line, e.g., accessed on 5 November 2019, [61]). Cells were cultured in Dulbecco’s Modified Eagle Medium with low glucose (Merck, Darmstadt, Germany) containing 1% non-essential amino acids (Merck, Germany), 15% fetal bovine serum (Merck, Darmstadt, Germany), and 1% penicillin/streptomycin (Merck, Germany). Cells were maintained in a humidified incubator at 5% CO_2_ and 37 °C.

### 4.2. Treatments and Irradiation

FaDu and UM-SCC-1 wt cells were treated with an ATRi (AZ20, 0.5 μM, Tocris, Minneapolis, MN, USA) or a Chk1i (MK-8776, 1 μM, Selleckchem, Houston, TX, USA) for at least 1 h before X-ray exposure or simultaneously with CDDP (Accord Healthcare GmbH, München, Germany). Cells were exposed to X-rays (140 kV) at room temperature using the D3150 X-ray Therapy System (Gulmay Medical Ltd., Byfleet, UK) at a dose rate of 3.62 Gy/min or were mock-treated (0 Gy), i.e., kept for the same time in the radiation device control room.

### 4.3. Clonogenic Survival Assay

HNSCC cells were seeded at low densities in 6-well plates. At 24 h after seeding, the cells were mock-treated or treated with the ATRi or Chk1i followed by exposure to 0, 2, 4, and 6 Gy X-rays or 0, 0.1, 0.25, 0.5, 1, and 2 µM CDDP. After another 24 h, CDDP was removed by sucking off the media. The cells were washed with PBS three times and supplied with fresh media without CDDP and the ATRi or Chk1i. The cells were incubated for 10–14 days to allow colony formation. Staining, evaluation, and data analysis were performed as described previously [62,63]. At least three independent experiments were performed with biological triplicates.

### 4.4. Cell Cycle Analysis

HNSCC cells were seeded in 6-well plates. After 48 h, exponentially growing cells were treated with the ATRi or Chk1i, mock-treated and exposed to 4 Gy X-rays, or treated with 2 µM CDDP. At 24 h after X-ray or CDDP exposure, cells were fixed, stained, and analyzed by flow cytometry as described previously [43,64]. At least three independent experiments with biological triplicates were performed.

### 4.5. Cytokinesis Block Micronucleus Assay

HNSCC cells were seeded in 6-well plates. After 48 h, exponentially growing cells were treated with the ATRi or Chk1i or mock-treated and exposed to 4 Gy X-rays or 2 µM CDDP. Cytochalasin B (4 μg/mL, Biomol, Hamburg, Germany) was added to abrogate cytokinesis and accumulate binucleated cells. At 48 h after X-ray exposure or 24 h after CDDP treatment, cells were harvested, treated for 10 min with 0.075 M potassium chloride solution (Merck, Germany) for hypotonic shock, and fixed twice with Carnoy’s solution. Cells were dropped on slides and mounted with a HOECHST33258 antifade mountant. Image acquisition by fluorescence microscopy, processing, and scoring of MN in binucleated cells was performed as described previously [65]. For each data point, MN were scored manually in at least 400 binucleated cells. At least 3 independent experiments were performed with biological triplicates.

### 4.6. Immunostaining

Immunostaining for protein expression of cGAS or cdsDNA and image acquisition by fluorescence microscopy was performed as described previously [43]. Briefly, HNSCC cells were seeded on glass coverslips in 6-well plates and treated according to the CBMN assay. At 24 h after exposure, cells were fixed in 3.7% formaldehyde/PBS for 10 min, permeabilized in 5% bovine serum albumin/0.5% Triton-X-100/PBS for 30 min at room temperature, and incubated with an anti-cGAS antibody (Cell Signaling Technology, Danvers, MA, USA). For the immunostaining of cdsDNA with an anti-dsDNA antibody (Abcam, Cambridge, UK), a mild permeabilization protocol was applied using 0.1% Tween-20 and 0.01% Triton-X-100 in PBS for 3 × 5 min at room temperature not affecting the nuclear envelope. Incubation with the primary antibody was performed overnight at 4 °C. Cells were washed in PBS and incubated with a secondary antibody conjugated with Alexa Fluor 488^®^ or Cy5 (Thermo Fisher Scientific, Dreieich, Germany) for 1 h at room temperature followed by mounting in an HOECHST33258-containing antifade mountant. Fluorescence microscopic image acquisition was performed with an Axio Imager.A1 microscope (Zeiss, Jena, Germany).

### 4.7. Western Blot

Exponentially growing HNSCC cells were treated with the ATRi or Chk1i or mock-treated and exposed to 2, 8, or 20 Gy X-rays or 2 µM CDDP and incubated up to 72 h. Lysis of cells, protein quantification, gel electrophoresis, Western blotting, and immune detection were performed as described previously [64]. Primary antibodies against the following target proteins were used: cGAS, phospho-STING(S366), STING, phospho-TBK1(S172), TBK1, phospho-STAT1(Tyr701), STAT1, phospho-IRF3(S396), IRF3, IFN-ß1, TREX1, PD-L1 (all obtained from Cell Signaling Technology, Danvers, MA, USA), and vinculin (Abcam, Cambridge, UK) followed by horseradish peroxidase (HRP)-conjugated goat IgG (Amersham™, Little Chalfont, UK). For multiple subsequent detections, the membranes were stripped using a mild stripping buffer (15 g/l Glycin, 1 g/L SDS, 10% Tween 20, pH 2.2) and reprobed. At least three independent experiments were performed.

### 4.8. Quantification of Cytosolic DNA

For the extraction of cdsDNA, fractional lysis of the cytosol from HNSCC cells was performed according to the protocol of Gagnon et al. [66]. Briefly, exponentially growing HNSCC cells were treated with the ATRi or Chk1i or mock-treated and exposed to 2, 8, and 20 Gy X-rays or 2 µM CDDP. After 72 h, cells were harvested and cell numbers were determined. A total 100 µL hypotonic lysis buffer (10 mM Tris pH 7.5, 10 mM NaCl, 3 mM MgCl_2_, 0.3% *vol*/*vol* NP-40, and 10% *vol*/*vol* glycerol) was added per 100,000 cells and lysis was conducted for 15 min on ice. The unlysed nuclei and cell debris were separated by centrifugation. cdsDNA in the supernatant was measured using a Pico488 dsDNA quantification kit (Lumiprobe, Germany) and a FLUOstar Omega microplate reader (BMG Labtech, Ortenberg, Germany). The concentration of cdsDNA was determined for each measurement utilizing a standard curve (Appendix A). At least three independent experiments were performed.

### 4.9. Data and Statistical Analysis

Data handling, plotting, and statistics were performed using SigmaPlot Version 14 (Systat Software, Richmond, CA, USA). For comparison of the means of two or more groups, Student’s t-test or one-way analysis of variance (ANOVA) was used, respectively. The local significance level was set to α = 0.05.

## Figures and Tables

**Figure 1 ijms-24-14900-f001:**
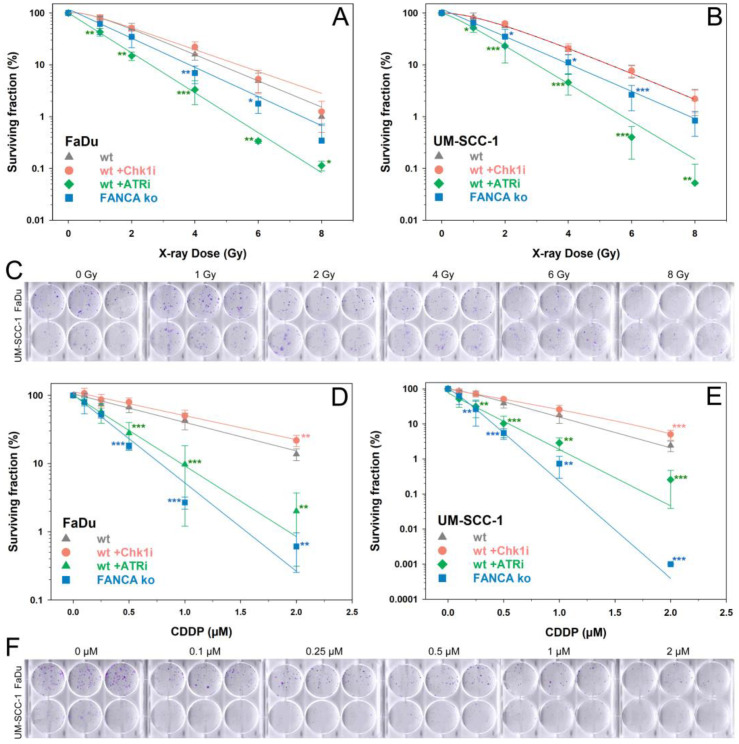
Clonogenic survival of FaDu cells (p53 mutated) or UM-SCC-1 cells (p53 non-mutated) after treatment with X-rays (**A**–**C**) or cisplatin (CDDP) (**D**–**F**) without or with inhibition of ATR, Chk1, or Fanconi anemia gene A knockout (FANCA ko). Representative images of stained colonies in 6-well plates are shown for FaDu and UM-SCC-1 wt cells after X-ray or CDDP treatment in C and F, respectively. Data show the means and standard deviations of at least 3 independent replicate experiments with biological triplicates. Statistically significant deviations from controls (wt) at any dose point were analyzed by one-way ANOVA: * *p* < 0.05, ** *p* ≤ 0.01, *** *p* ≤ 0.001. All lines were fitted by a linear quadratic fit.

**Figure 2 ijms-24-14900-f002:**
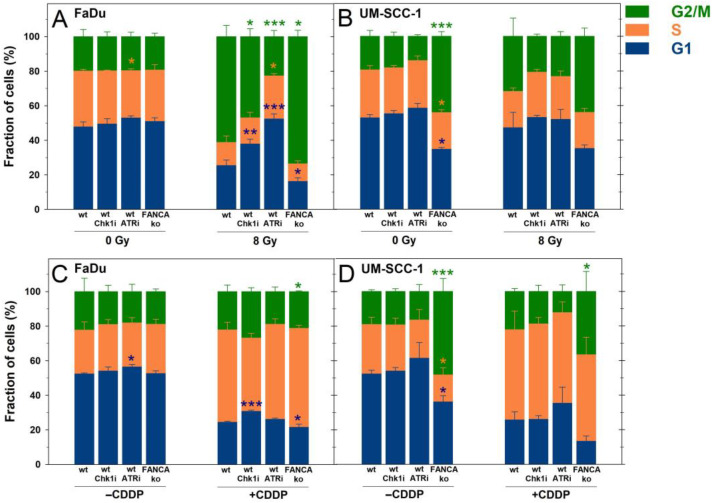
Cell cycle distributions of FaDu cells (left panel) or UM-SCC-1 cells (right panel) 24 h after treatment with 8 Gy X-rays (**A**,**B**) or 2 µM cisplatin (CDDP), (**C**,**D**) without or with inhibition of ATR, Chk1, or Fanconi anemia gene A knockout (FANCA ko). Data show the means and standard deviations of at least 3 independent replicate experiments with biological triplicates. Statistically significant deviations from controls (wt) for each treatment were analyzed by one-way ANOVA: * *p* < 0.05, ** *p* ≤ 0.01, *** *p* ≤ 0.001.

**Figure 3 ijms-24-14900-f003:**
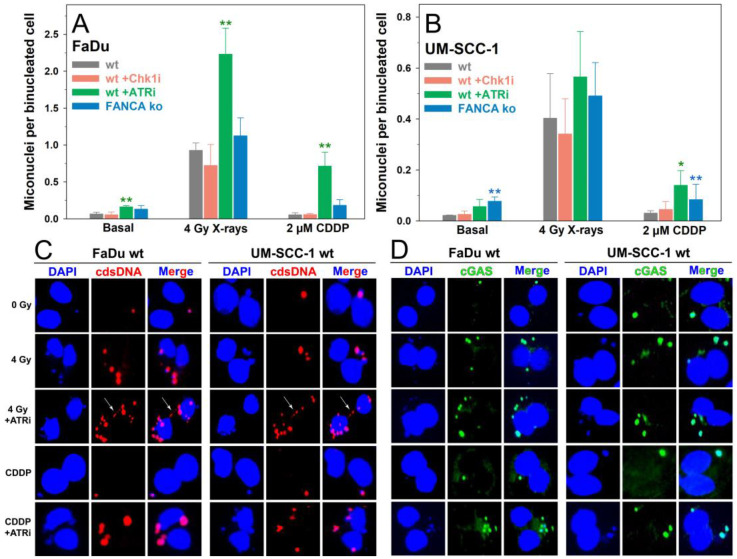
Micronuclei in binucleated FaDu (**A**) or UM-SCC-1 cells (**B**) after treatment with 4 Gy X-rays or 2 µM cisplatin (CDDP) without or with inhibition of ATR, Chk1, or Fanconi anemia gene A knockout (FANCA ko). Data show the means and standard deviations of at least 3 independent replicate experiments with biological triplicates. Statistically significant deviations from control for each treatment (wt) were analyzed by one-way ANOVA: * *p* < 0.05, ** *p* ≤ 0.01. Representative immunofluorescence images after immunostaining for cytosolic double-stranded DNA (cdsDNA) (**C**) or cGAS (**D**) in binucleated FaDu and UM-SCC-1 wt cells after indicated treatments. For the detection of cdsDNA with a monoclonal antibody, a mild permeabilization protocol not affecting the nuclear envelope was applied. For the immunolabeling of cGAS, a conventional permeabilization protocol was applied. Positive staining signals were restricted to MN and chromatin bridges (white arrows) for cdsDNA (**C**) and to MN for cGAS (**D**).

**Figure 4 ijms-24-14900-f004:**
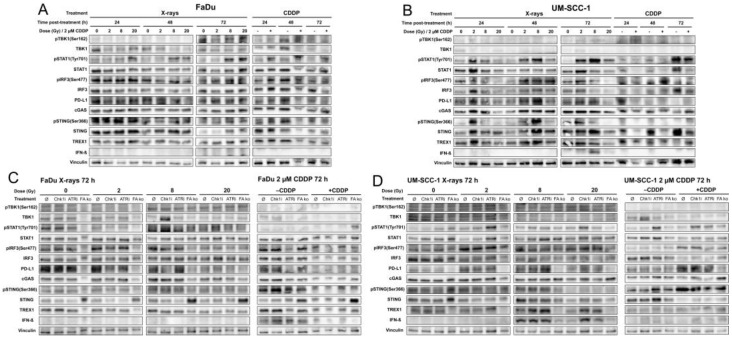
Exemplary Western blots of FaDu (**A**,**C**) and UM-SCC-1 (**B**,**D**) cells 24, 48, and 72 h after exposure to 0, 2, 8, or 20 Gy X-rays or 2 µM cisplatin (CDDP) (**A**) and alone (**B**) without or with inhibition of ATR, Chk1, or Fanconi anemia gene A knockout (FANCA ko) 72 h after the start of treatments (**C**,**D**). A single experiment was performed for the results shown in A and B, and three independent replicate experiments were performed for the results shown in (**C**,**D**); each of these are provided in Appendix A. Densitometric quantification of signals is provided in Appendix A.

**Figure 5 ijms-24-14900-f005:**
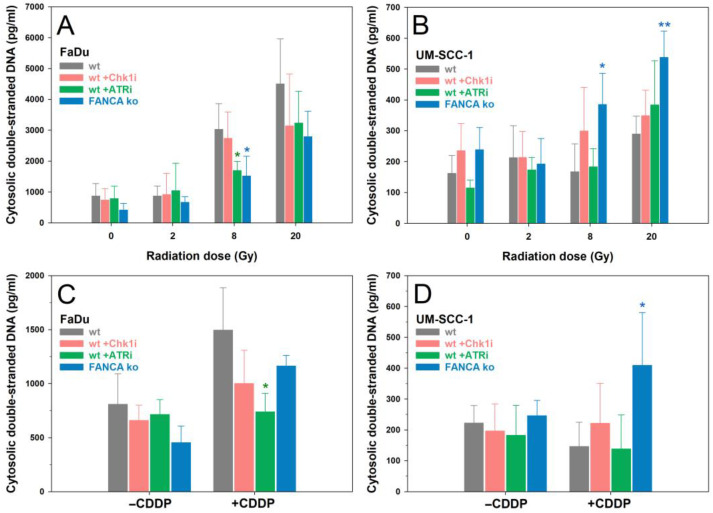
Quantification of cytosolic double-stranded DNA in FaDu (**left panel**) and UM-SCC-1 (**right panel**) cells 72 h after exposure to 0, 2, 8, or 20 Gy X-rays (**A**,**B**) or 2 µM cisplatin (CDDP) (**C**,**D**) without or with inhibition of ATR, Chk11, or Fanconi anemia gene A knockout (FANCA ko). Data show the means and standard deviations of at least 3 independent replicate experiments. Statistically significant deviations from control (wt) for each treatment condition were analyzed by one-way ANOVA: * *p* < 0.05, ** *p* ≤ 0.01.

## Data Availability

All data are available upon request.

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
