# Peer review of "The cGAS/STING/IFN-1 Response in Squamous Head and Neck Cancer Cells after Genotoxic Challenges and Abrogation of the ATR-Chk1 and Fanconi Anemia Axis"

_ijms, 2023, doi:10.3390/ijms241914900_

Round 1

Reviewer 1 Report

This article proposes to investigate the role of the cGSAS/STING/IFN-1 signaling pathway in the induction of tumor immune responses following DNA-damaging anticancer therapies such as CDDP or IR in monotherapy or combination with pharmacological inhibitors of the ATR or FA pathways in wild-type or FA pathway-deficient HNSCC cell lines.

The article is well presented, is novel, and may be of interest in the search for new therapeutic strategies in this type of neoplasm.

Questions for the authors

-What is the percentage of patients with HNSCCs with compromised FA pathway?

-Regarding the treatments used in the study, why were low and high doses of radiation used? In general, low doses induce DNA damage but not cell death. Did the authors study the levels of cell death induced by both doses of radiation in their cell line model??

-Regarding the dose of CDDP, why did they choose 2 uM? Did the authors test the effect of this dose on cell death?

-It would be interesting to explain the concepts of hypofractionated or stereotactic multimodal RT-ICI for HNSCC in the introduction. It would help to understand better the importance of the study in the search for new therapeutic combinations with different doses of radiation.

Fig 1. In clonogenicity assays, have the authors tested whether these same experimental conditions induce cell death?

Fig 4. In this figure, some of the proteins studied are increased in favor of loading, as observed by the vinculin measurement. It would be desirable to quantify the protein expression results to have more precise data about the proteins that are modified after the different treatments.

Author Response

We thank the reviewers for their effort, time, and helpful comments. We are grateful for the opportunity to address these concerns and doubts and have revised the manuscript accordingly. The attached PDF file contains the detailed point-by-point response to the comments. 

Reviewer 2 Report

In this study, authors assessed the cGSAS/STING/IFN-1 signaling pathway after X-ray or CDDP exposure in HNSCC cells with or without ATR- or Chk1 suppression as well as a FANCA knock-out model. The authors have used a variety of phenotypical and molecular approaches to assess the cytotoxic effect, cell cycle arrest, changes in gene expression, and micronuclei formation in various DNA damaging experimental setups. In conclusion authors suggested importance of cGSAS/STING/IFN-1-mediated antitumor immune response in HNSCC by DNA damage inducing drug treatment or radiation.  

Overall, the study is original, well-designed, and uses the right experimental methodology. However, some high-toned statements can be less accurate in the absence of a microenvironment or in-vivo system. There was neither normal nor non-tumor control.

Other- minor comments:

1.      The abbreviations should be addressed properly.

2.      Authors should provide representative images for Figure 1.

3.      Western blot images should be quantiated and molecular weight markers should be included.

Author Response

(The authors gave the same response as above.)

Reviewer 3 Report

Here the authors present an article that reports the results of a study on the effects of different treatments on the cGAS/STING/IFN-1 pathway in head and neck squamous cell carcinoma (HNSCC) cells. The authors investigated how exposure to X-rays or cisplatin, and inhibition of ATR, Chk1, or FANCA genes, affected the expression and activity of this pathway, which is involved in the antitumor immune response. They found that the response was variable depending on the cell line, the treatment, and the dose. They suggested that hypofractionated or stereotactic radiotherapy combined with immuno-oncological strategies could enhance the cGAS/STING/IFN-1-mediated antitumor immune response in HNSCC1.

Main points: 

The article is well-structured, with clear sections, figures, and references. The methods are detailed and the results are supported by statistical analysis. 

The article addresses a clinically relevant topic and provides novel insights into the molecular mechanisms of HNSCC response to genotoxic treatments and immunotherapy.

Minor points: 

The article has some limitations, such as the use of only two HNSCC cell lines, which may not represent the heterogeneity of this tumor type. 

The article also lacks in vivo validation of the findings (for ex, xenografts) and does not explore the effects of other DNA damage response inhibitors or immune checkpoint inhibitors. 

Author Response

(The authors gave the same response as above.)

Round 2

Reviewer 3 Report

The authors addressed all concerns.